# Biological Activities of Ethanol Extracts of *Hericium erinaceus* Obtained as a Result of Optimization Analysis

**DOI:** 10.3390/foods13101560

**Published:** 2024-05-16

**Authors:** Mustafa Sevindik, Ayşenur Gürgen, Vadim Tagirovich Khassanov, Celal Bal

**Affiliations:** 1Department of Biology, Engineering and Natural Sciences Faculty, Osmaniye Korkut Ata University, 80000 Osmaniye, Türkiye; 2Department of Industrial Engineering, Engineering and Natural Sciences Faculty, Osmaniye Korkut Ata University, 80000 Osmaniye, Türkiye; aysenurgurgen@osmaniye.edu.tr; 3Department of Biology, Agronomic Faculty, Saken Seifullin Kazakh Agrotechnical University, Astana 010011, Kazakhstan; vadim_kazgatu@mail.ru; 4Department of Biology, Science and Literature Faculty, Gaziantep University, 27310 Gaziantep, Türkiye; bal@gantep.edu.tr

**Keywords:** anti-Alzheimer activity, anticancer activity, antimicrobial activity, antioxidant activity, *Hericium erinaceus*, medicinal mushroom, optimization

## Abstract

Mushrooms are one of the indispensable elements of human diets. Edible mushrooms stand out with their aroma and nutritional properties. In this study, some biological activities of the wild edible mushroom *Hericium erinaceus* were determined. In this context, firstly, the most suitable extraction conditions of the fungus in terms of biological activity were determined. First, 64 different experiments were performed with the Soxhlet device under 40–70 °C extraction temperature, 3–9 h extraction time, and 0.5–2 mg/mL extraction conditions. As a result, a total antioxidant status (TAS) analysis was performed, and the extraction conditions were optimized so that the objective function was the maximum TAS value. The data obtained from the experimental study were modeled with artificial neural networks (ANNs), one of the artificial intelligence methods, and optimized with a genetic algorithm (GA). All subsequent tests were performed using the extract obtained under optimum extraction conditions. The antioxidant capacity of the mushroom was assessed using Rel assay kits and the DPPH and FRAP techniques. The agar dilution method was used to measure the antimicrobial activity. The anti-Alzheimer activity was assessed based on the activities of acetylcholinesterase (AChE) and butyrylcholinesterase (BChE). The antiproliferative activity was assessed against the A549 cancer cell line. The total phenolic content was measured using the Folin–Ciocalteu reagent. The measurement of total flavonoids was conducted using the aluminum chloride test. LC-MS/MS equipment was used to screen for the presence of standard chemicals. The optimum extraction conditions were found to be a 60.667 °C temperature, 7.833 h, and 1.98 mg/mL. It was determined that the mushroom has high antioxidant potential. It was determined that the substance was successful at combating common bacterial and fungal strains when used at dosages ranging from 25 to 200 µg/mL. The high antiproliferative effect of the substance was attributed to its heightened concentration. The anti-AChE value was found to be 13.85 μg/mL, while the anti-BChE value was confirmed to be 28.00 μg/mL. The phenolic analysis of the mushroom revealed the presence of 13 chemicals. This investigation found that *H. erinaceus* exhibits robust biological activity when extracted under appropriate circumstances.

## 1. Introduction

Mushrooms are natural products that stand out with their different features. It is known that mushrooms, which occupy an important place in people’s diets, also have high medical potential. Mushrooms have been found to consist of 57% carbohydrates (with 8% being dietary fiber), 4% fat, and 22% protein [1]. Many studies have demonstrated that many mushroom species exhibit distinct biological activities, including anticancer, antiproliferative, antioxidant, antibacterial, and DNA-protecting properties [2,3,4].

*Hericium erinaceus* (lion’s mane mushroom, mountain priest mushroom, bearded tooth mushroom) has a cosmopolitan distribution and is found on hard trees in the form of a cluster of hanging thorns arising from a single root. It is especially common in the autumn months [5]. It is known that it is widely used in Chinese medicine. In this context, it is produced on logs in Far East Asian countries. There have been numerous types of biologically active chemicals derived from *H. erinsaceus* that have been proven to possess pharmacological activity and therapeutic characteristics [6,7,8]. It has been proven by previous studies that *H. erinsaceus* has bioactive properties, such as antioxidant, anti-inflammatory, antiallergic, antimicrobial, antibofilm, anti-aging, and neuroprotective properties [8,9,10,11]. It has also been proven that these bioactive properties are also affected by the extraction conditions of the *H. erinsaceus*, such as the solvent used in the preparation of the extract [12]. As a result, optimizing the extraction parameters is one of the approaches to achieve the best possible yield from *H. erinsaceus*. In a previous study, the antioxidant potential of *H. erinaceus* extracts obtained through ultrasonic extraction with ethanol was investigated. The extraction process was optimized through the use of response surface methodology (RSM) by varying the parameters in order to determine the optimal extraction yield of metabolites that were involved in the antioxidant activity that was being investigated. It has been reported that the optimal extraction conditions are a solvent-to-material ratio of 1:30 (g/mL), an extraction period of 45 min, and an ethanol concentration of 80% [13].

Recent research has shown that artificial intelligence techniques are commonly employed for extract optimization due to complicated systems. This is due to the fact that these strategies are both reliable and effective, and they have an outstanding ability to describe the non-linear interactions that exist between the parameters and the responses of various systems [14,15].

This study became more informative and significant by optimizing the extraction conditions and subsequently examining and characterizing the bioactive properties of the produced extracts. This is the first study to optimize extracts using artificial intelligence techniques. In this study, we aimed to determine the chemical and biological characterization of extracts of *H. erinaceus* mushroom obtained under optimum conditions using artificial neural networks and a genetic algorithm.

## 2. Materials and Methods

The mushroom specimens included in the investigation were gathered from Trabzon, Turkey. Fungal samples were collected from *Quercus* sp. In addition, young samples of the mushroom were preferred. Following the identification procedures, the most suitable extraction conditions for the mushroom samples were determined.

### 2.1. Extraction Procedure Method

Firstly, 64 different experiments (4 × 4 × 4) were carried out in the Gerhardt SOX-414 (Königswinter, Germany) device using ethanol as the solvent, including 40, 50, 60, and 70 °C extraction temperatures, 3, 5, 7, and 9 h extraction times, and 0.5, 1.0, 1.5, and 2 mg/mL extraction conditions. All experiments were performed in 3 replicates and the mean and standard deviation of each group were calculated. The obtained data were modeled with ANNs and optimized with a GA.

### 2.2. Modeling

Modeling was performed with the ANN method. While the inputs of the model were the extraction temperature, extraction time, and extract concentration, the output was used as the TAS value. The layers of the ANN used in this study are given in Figure 1.

Of the data obtained from the experimental studies, 80% were used for training, 10% were used for validation, and 10% were used for testing. The Levenberg–Marquardt (LM) algorithm was used for the learning process. The most appropriate network was determined by comparing the numbers of 20 different (1:20) hidden neurons. The learning coefficient and momentum coefficient were chosen as 0.5, the maximum number of iterations was 500, the number of verification checks was 50, and the error value was 1 × 10^−5^. Since there is a possibility of the performance surface becoming stuck in a local minimum during the training of the network, a single training run may not produce the optimum performance. Therefore, in order to approach a global minimum value, the networks must be retrained several times to obtain the optimal network. In this study, a total of 1000 training sessions were performed for each model. In the study, the mean square error (MSE) and mean absolute percentage error (MAPE) were used as the performance indicators of the developed models. The MSE and MAPE were calculated according to Equations (1) and (2), as follows: (1)$$\mathrm{MSE}=\;\frac{1}{n}\overset{n}{\underset{i=1}{\sum }}{\left(e_{i}-p_{i}\right)}^{2}\;$$
(2)$$\mathrm{MAPE}=\frac{1}{n}\sum \left|\frac{e_{i}-p_{i}}{e_{i}}\right|*100\;$$
where  e is the experimental result, p  is the prediction result, and n is the number of samples.

### 2.3. Optimization

The optimization process was conducted using a genetic algorithm (GA). Studies have been carried out for different population numbers and the roulette-wheel technique has been used for natural selection. For crossover, the single-point crossover method was used. The appropriate number of iterations was determined by analyzing the convergence graphs. Each optimization study was repeated at least 60 times, producing results very close to the global optimum.

### 2.4. Extraction Processes

In the optimization study conducted to determine the extraction conditions that maximize the biological activity of mushroom samples, the optimum extraction conditions were determined (60.667 °C temperature, 7.833 h, and 1.98 mg/mL). Then, extracts were obtained by adjusting the optimum extraction conditions (61 °C temperature, 7 h 50 min, and 2 mg/mL) in the computer environment with the Gerhardt SOX-414 device. All analyses in the study were made on extracts obtained under the optimum extraction conditions. After the extraction process, ethanol, used as solvent, was evaporated using a Buchi R100 Rotary Evaporator (Buchi, R100, Flawil, Switzerland) and crude extracts were obtained.

### 2.5. MTT Assay

The inhibitory effect on the cell growth of the mushroom extract was assessed using the MTT (3-(4,5-Dimethylthiazol-2-yl)-2,5-diphenyltetrazolium bromide) test against the A549 lung cancer cell line. The solutions were produced using the extracts at concentrations of 25, 50, 100, and 200 μg/mL. Subsequently, a cell confluence of 70–80% was attained, and the separation process was carried out using a 3.0 mL solution of trypsin–EDTA (Sigma-Aldrich, St. Louis, MO, USA). Subsequently, the plates were inoculated and placed in an incubator for a duration of 24 h. Subsequently, stock solutions were introduced and subjected to a 24 h incubation period. The supernatants were subsequently mixed with the growth medium and substituted with a solution of MTT at a concentration of 1 mg/mL. The sample was kept at a temperature of 37 °C until a purple precipitate was produced. Next, dimethyl sulfoxide (DMSO) from Sigma-Aldrich in MO, USA, was introduced to the MTT solution. The absorbance reading was then taken at a wavelength of 570 nm using an Epoch spectrophotometer manufactured by BioTek Instruments in Winooska, Vermont [16].

### 2.6. Anticholinesterase Analysis

The anticholinesterase activity of the mushroom extract was determined by modifying the Ellman method [17]. Galantamine was employed as the established benchmark. Stock solutions were created using mushroom extracts at doses ranging from 200 to 3.125 μg/mL. Next, 130 microliters of a 0.1 molar phosphate buffer with a pH of 8, 10 microliters of the stock solution, and 20 microliters of either AChE (5.32 × 10^−3^ U) or BChE (6.85 × 10^−3^ U) enzyme solution were introduced into the microplate. Subsequently, the sample was subjected to incubation for a duration of 10 min at a temperature of 25 °C, while being kept in a light-free environment. Next, 20 μL of DTNB (5,5″-dithiobis-(2-nitrobenzoic acid)) solution and 20 μL of either acetylcholine iodide (0.71 mM) or butyrylcholine iodide (0.2 mM) substrate were introduced. The measurement was then taken at a wavelength of 412 nm. The samples underwent three repetitions. The IC50 values, representing the percentage of inhibition of the samples, were given in terms of micrograms per milliliter (μg/mL).

### 2.7. Antimicrobial Analysis

The antimicrobial activity of the mushroom extract against the test bacterial and fungus strains was assessed using the agar dilution method. The test bacteria were grown in Hinton Broth medium prior to the experiment. The fungi used for testing were grown in RPMI 1640 Broth medium prior to the experiment. The results indicated the minimum concentration of extract required to inhibit bacterial growth. The antimicrobial activity of the mushroom extract was evaluated at concentrations ranging from 12.5 to 800 µg/mL [18,19,20,21].

Test bacteria: *Staphylococcus aureus* ATCC 29213; *S. aureus* MRSA ATCC 43300; *Enterococcus faecalis* ATCC 29212; *Escherichia coli* ATCC 25922; *Pseudomonas aeruginosa* ATCC 27853; and *Acinetobacter baumannii* ATCC 19606.

Test fungi: *Candida albicans* ATCC 10231; *C. krusei* ATCC 34135; and *C. glabrata* ATCC 90030.

### 2.8. Total Phenolic Analysis

Stock solutions of 1 mL were made using the mushroom extract, and these solutions were supplemented with 1 mL of Folin–Ciocalteu reagent in a ratio of 1:9 (*v*/*v*). Next, the solution was vigorously mixed and 0.75 mL of a 1% Na_2_CO_3_ solution was introduced. The sample was thereafter placed in an incubator at ambient temperature for a duration of 2 h, followed by measurement at a wavelength of 760 nm. The total phenolic content was quantified in milligrams per gram using the calibration curve of the gallic acid standard solution [22].

### 2.9. Antioxidant Analysis

#### 2.9.1. Total Antioxidant Status and Total Oxidant Status

The mushroom extract was assessed for its total antioxidant status (TAS) and total oxidant status (TOS) using Rel Assay TAS and TOS kits (Rel Assay, Megatıp, Şehitkamil/Gaziantep, Türkiye). The analysis was conducted in accordance with the manufacturer’s indicated procedure. The TAS values were quantified in terms of millimoles of Trolox equivalent per liter, whereas the TOS values were measured in terms of micromoles of hydrogen peroxide equivalent per liter [23,24]. The oxidative stress index (OSI) value was calculated by comparing the percentages of the TOS values and TAS values [25].

#### 2.9.2. DPPH Free Radical Scavenging Activity

Stock solutions with a concentration of 1 mg/mL were produced from the mushroom extract using DMSO. One milliliter of the solution was added to 160 microliters of DPPH solution, which had a concentration of 0.267 millimolars and a volume of 4 milliliters. The DPPH solution was prepared in a 0.004% methanol solution. Subsequently, the sample was subjected to incubation for a duration of 30 min under light-restricted conditions and at ambient temperature. The absorbance of the sample was then measured at a wavelength of 517 nm. The values are quantified in milligrams of Trolox equivalent per gram of extract [26].

#### 2.9.3. Ferric-Reducing Antioxidant Power Assay

A 100 µL stock solution was created using the mushroom extract. Next, the stock solution was combined with 2 mL of FRAP reagent, 300 mM acetate buffer (pH 3.6), 40 mM HCl, and 20 mM FeCl_3_ 6H_2_O solution, along with 10 mM 2,4,6-tris(2-pyridyl)-S-triazine solution. This mixture was then added to the produced FRAP solution in a ratio of 10:1:1. The sample was thereafter placed in an incubator set at a temperature of 37 °C for a duration of 4 min. Subsequently, a measurement was conducted at a wavelength of 593 nm. The values were denoted as milligrams of Trolox equivalent per gram of extract [26].

### 2.10. Phenolic Analysis

Within this study, we utilized the liquid chromatography–tandem mass spectrometry (LC-MS/MS) instrument to screen 24 typical chemicals within the fungus. The analysis was conducted using a C-18 Intersil ODS-4 (3.0 mm × 100 mm, 2 μm) analytical column with a column temperature of 40 °C. The mobile phase consisted of two components: A, which was a mixture of water and 0.1% formic acid, and B, which was a mixture of ethanol and 0.1% formic acid. The flow rate of the mobile phase was 0.3 mL/min, and the volume of the sample injected was 2 μL.

### 2.11. Statistical Analysis

The “SPSS 21.0 for Windows” program was used for the statistical analysis of all the analyses carried out within the scope of this study. Simple Variance Analysis (SVA) was performed to determine the difference between the groups in the tests studied; the Duncan test was applied at the confidence level α = 0.05 to determine the difference between the groups.

## 3. Results and Discussion

### 3.1. Optimization of Extraction Conditions

The optimization of the mushroom extraction conditions was carried out in three stages. These were the experimental-study, modeling, and optimization stages. Components with antioxidant activity are known to have other medicinal properties [27]. Therefore, the extraction conditions were optimized to maximize the total antioxidant potential. The total antioxidant status values of the ethanolic extracts of *H. erinaceus* are given in Table 1.

When Table 1 is examined, it is seen that the extraction conditions that give the lowest TAS value (2.038 ± 0.096) are 70 °C, 3 h, and a 0.25 mg/mL concentration. Among the studied limits, it was observed that the highest TAS value was 5.323 ± 0.109 under the 60 °C, 7 h, and 2 mg/mL extraction concentration conditions. As the concentration increased at all temperature and time values, the TAS values also increased. The TAS values decreased significantly when the extraction temperature increased to 70 °C. The data obtained were modeled with ANNs in order to estimate the intermediate values that were not made within the experimental limits and to carry out an optimization study. Among the established models, the architecture of the best prediction model was found to be 3-5-1. In other words, the prediction model produced using five hidden neurons was chosen as the best model. The MSE, MAPE, and R values of this model were calculated as 0.005, 1.499%, and 0.996, respectively. A regression plot of the TAS output is given in Figure 2. Also, the real values and ANN results are given in Figure 3.

The optimization process was carried out with the genetic algorithm method using the best selected ANN model. Among the different population numbers tried, the most appropriate population number was determined to be 20. After the optimization process of 60 iterations, the convergence curve of the optimization process was drawn (Figure 4), and it was seen that the objective-function value remained constant after the ninth iteration.

After the optimization process, the optimum extraction conditions were found to be a 60.667 °C temperature, 7.833 h, and 1.98 mg/mL. It was predicted that the maximum TAS value would be 5.5526 mmol/L under these extraction conditions.

### 3.2. Antiproliferative Activity

The increase in cancer cases in recent years has led researchers to investigate new treatment methods. The use of food supplements is recommended in cancer treatments, especially to strengthen the immune system [28]. In this study, the effect of *H. erinaceus* mushroom against the A549 cancer cell line was investigated. The findings obtained are shown in Figure 5.

A study conducted in China found that water and ethanol extracts of *H. erinaceus* exhibited antiproliferative action on the HepG2, Huh-7, HT-29, and NCI-87 cell lines [29]. A further study carried out in Turkey found that various extracts of *H. erinaceus*, including ethanol–water, ethanol, methanol–water, water, ethyl acetate, and ether extracts, had antiproliferative effects on MCF-7 cells [29]. We utilized ethanol extracts of *H. erinaceus* in this research to evaluate their efficacy against the A549 cancer cell line. The data revealed that the mushroom extract displayed potent activities that were directly correlated with the concentration level. It was shown that it displayed the greatest level of activity when the concentration was 200 μg/mL. Within this particular framework, it has been ascertained that mushrooms possess significant potential as anticancer agents.

### 3.3. Anti-Alzheimer Activity

There are many diseases caused by oxidative stress. Among these diseases, Alzheimer’s disease, which shows a significant increase with age, is seen to be especially common in people over the age of 65. It is thought that the number of cases may exceed 80 million in the near future. There is currently no known treatment for this disease, which occurs for different reasons. However, the use of food supplements that reduce the effect of oxidative stress may play a role in reducing the disease [30]. In this study, the anticholinesterase activity was detected using ethanol extracts of *H. erinaceus*. The IC50 values of the findings are shown in Table 2.

The literature does not contain any previous reports on the anticholinesterase activity of *H. erinaceus*. The activities of acetyl and butyrylcholinesterase in many fungus species have been documented [31,32,33]. We determined the activities of acetyl and butyrylcholinesterase in *H. erinaceus* as part of this investigation. Both the acetyl and butyrylcholinesterase activities were shown to be elevated compared to the conventional study of galantamine. The suppression of the enzymes that contribute to the development of human diseases can play a crucial role in combating diseases [34]. This investigation found that the mushroom extract utilized in this context has the potential to be employed in the treatment of neurodegenerative illnesses via its effective inhibition of cholinesterase enzymes.

### 3.4. Antimicrobial Potential

Currently, numerous antimicrobial medications employed to combat microbial illnesses are insufficient. The primary cause of this phenomenon is attributed to the proliferation of drug-resistant bacteria resulting from inadvertent drug consumption. Furthermore, the potential adverse effects of synthetic pharmaceuticals have prompted researchers to uncover novel antibacterial medications [35]. This study aimed to assess the antibacterial efficacy of *H. erinaceus* extract. The results are presented in Table 3.

A study conducted in Korea found that the ethanol extract of *H. erinaceus* demonstrated effectiveness against various bacteria, including *Listeria monocytogenes*, *Staphylococcus aureus*, *Pseudomonas fluorescens*, *Salmonella enteritidis*, *Vibrio parahaemolyticus*, and *Escherichia coli*. The extract was found to be effective at doses ranging from 1 to 10 mg/mL [36]. Another study found that extracts of *H. erinaceus* in water, methanol, and water–methanol were successful at combating *Pseudomonas aeruginosa* and *L. monocytogenes* [37]. In contrast to previous investigations, this study found that the extract of *H. erinaceus* prepared under optimal circumstances showed the greatest activity against *S. aureus* and *S. aureus* MRSA at a concentration of 25 µg/mL. Furthermore, it was found that it suppressed the proliferation of *E. faecalis* and *C. albicans* when present at a dose of 50 µg/mL. Furthermore, it was discovered that at a concentration of 100 µg/mL, it exhibited efficacy against *E. coli*, *P. aeruginosa*, *A. baumannii*, *C. glabrata*, and *C. krusei*. Within this particular framework, it was ascertained that the mushroom extract employed in this investigation exhibited a highly potent antibacterial capacity within the concentration range of 25–100 µg/mL. Consequently, it was concluded that the mushroom extract exhibited significant antibacterial properties.

### 3.5. Total Phenolic Contents

Various bioactive chemicals are accountable for several biological actions. We measured the total phenolic content of the ethanol extract of *H. erinaceus* in this investigation. The findings (total antioxidant, total oxidant, oxidative stress index, and total phenolic values) obtained are shown in Table 4.

Previous reports have indicated that the total phenolic contents of water, methanol, and water–methanol extracts of *H. erinaceus* vary from 1.96 to 6.31 mg/g [37]. A separate investigation found that n-hexane, chloroform, ethyl acetate, n-butanol, and water extracts of *H. erinaceus* had concentrations ranging from 4.36 to 35.18 mg/g [38]. A further investigation found that the ethanol extract of *H. erinaceus* had a total phenolic content of 41.28 mg/g [39]. The study utilized an ethanol extract of *H. erinaceus*, and it was shown that the extracts produced under ideal conditions had a concentration of 59.75 ± 1.82 mg/g. It was found to have a greater total phenolic content compared to those of other extracts mentioned in the literature. Within this context, it has been established that it can serve as a significant reservoir in relation to the overall phenolic content.

### 3.6. Antioxidant Activity

Free radicals are reactive molecules that are generated as a byproduct of metabolic processes. Elevated concentrations of these substances can result in cellular harm. Antioxidants serve to diminish or inhibit the impacts of these substances [40]. Oxidative stress arises from the disparity between the levels of antioxidants and oxidant chemicals. Oxidative stress can lead to the development of severe ailments in humans, including cancer, cardiological illnesses, and neurological diseases. Additional antioxidants can help to prevent certain disorders [41]. This study aimed to investigate the antioxidant capacity of *H. erinaceus* in the given situation. The results are presented in Table 4.

The TAS, TOS, and OSI values of *H. erinaceus* have not been reported in the literature. They were detected for the first time in this study. Additionally, the FRAP activities of n-hexane, chloroform, n-butanol, and water extracts of *H. erinaceus* were previously reported as 10.66–174.82 µmol/g. In the same study, it was reported that the DPPH activities of the extracts had 16.83–71.70% inhibition at 1.5 mg/mL [38]. In this study, the DPPH value of the ethanol extract of *H. erinaceus* was determined as 73.36 ± 2.04 mg Trolox equivalent/g, and the FRAP value was 107.66 ± 2.41 mg Trolox equivalent/g. The literature has investigations on various wild mushrooms, including on the TAS, TOS, and OSI. In one of these studies, *Agrocybe praecox* mushroom had a TAS value of 2.97 mmol/L, a TOS value of 7.63 µmol/L, and an OSI value of 0.26 [42]. The TAS value of *Candolleomyces candolleanus* has been reported as 5.547 mmol/L, the TOS value as 8.572 µmol/L, and the OSI value as 0.155 [43]. The TAS value of *Entoloma sinuatum* was recorded as 2.64 mmol/L, the TOS value as 6.58 µmol/L, and the OSI value as 0.25 [44]. In this investigation, we observed that the TAS value of *H. erinaceus* was higher than those of *A. praecox* and *E. sinuatum* but lower than that of *C. candolleanus*. The TAS value quantifies the collective amount of antioxidant chemicals generated by the mushroom. Elevated TAS readings suggest that the mushroom possesses a significant amount of antioxidants. Based on the findings from this study, the extracts of *H. erinaceus* prepared under ideal conditions showed significant antioxidant capabilities. The TOS value is a measure of the overall amount of oxidant chemicals generated in the mushroom. The TOS value of *H. erinaceus* utilized in this research was found to be lower than those of *A. praecox* and *C. candolleanus* and greater than that of *E. sinuatum*. Based on the findings from this investigation using *H. erinaceus*, it was established that the quantities of oxidant compounds in the extracts produced under optimal conditions were within the normal range. The OSI value is calculated by dividing the mushroom’s TOS value by its TAS value and expressing it as a percentage. The OSI number represents the proportion of antioxidant chemicals that inhibit oxidant substances. The *H. erinaceus* specimens used in this investigation exhibited lower OSI values compared to those of *A. praecox*, *C. candolleanus*, and *E. sinuatum*. Within this particular context, it has been noted that *H. erinaceus* demonstrates a noteworthy capacity for inhibiting oxidant chemicals. Consequently, it seems that extracts of *H. erinaceus* acquired under ideal circumstances have significant antioxidant activity.

### 3.7. Phenolic Contents

Fungi produce many biologically active secondary metabolites. While these compounds are not nutritionally important, they have very important medical potential. In this study, standard compounds found in *H. erinaceus* were scanned on the LC-MS/MS device. The findings obtained are shown in Table 5.

The examination revealed the presence of acetohydroxamic acid, catechinhyrate, resveratrol, myricetin, fumaric acid, gallic acid, protocatechuic acid, 4-hydroxybenzoic acid, phloridzindyhrate, 2-hydoxycinamic acid, naringenin, quercetin, and luteolin in *H. erinaceus*. The literature reports the presence of 3,4 dihydroxy-benzoic acid, caffeic acid, syringic acid, routine, ellagic acid, *p*-coumaric acid, salicylic acid, vanillin, ferulic acid, sinapic acid, rosmarinic acid, and t-cinnamic acid in *H. erinaceus* [45]. A separate investigation documented the existence of 4-hydroxybenzoic acid, α-resorcylic acid, 4-coumaric acid, ferulic acid, and syringic acid in *H. erinaceus* [38]. A further investigation revealed the existence of protocatechuic acid, *p*-coumaric acid, succinic acid, catechin, 2-hydroxybenzoic acid, and ferulic acid in *H. erinaceus* [11]. Within this particular setting, this work revealed that *H. erinaceus* could serve as a significant reservoir of phenolic chemicals. Furthermore, it is believed that the discrepancies observed in these mushrooms as described in the literature are attributed to the many stressors experienced by the mushroom in its habitat and the variations in the geographical locations from where it is gathered. Therefore, it is believed that *H. erinaceus* could serve as a natural reservoir of pharmacologically active chemicals.

## 4. Conclusions

This study aimed to assess the biological activity derived under the optimal extraction conditions of extracts of the wild edible mushroom *H. erinaceus*. According to the findings, it was determined that extracts obtained in this way have high antioxidant, antiproliferative, and antimicrobial potential. In addition, 13 phenolic compounds were detected in the mushroom. By comparing our data with data from the literature, it was determined that the optimally produced extracts showed higher activities. As a result, it can be seen that *H. erinaceus* is an important natural resource for pharmacological designs.

## Figures and Tables

**Figure 1 foods-13-01560-f001:**
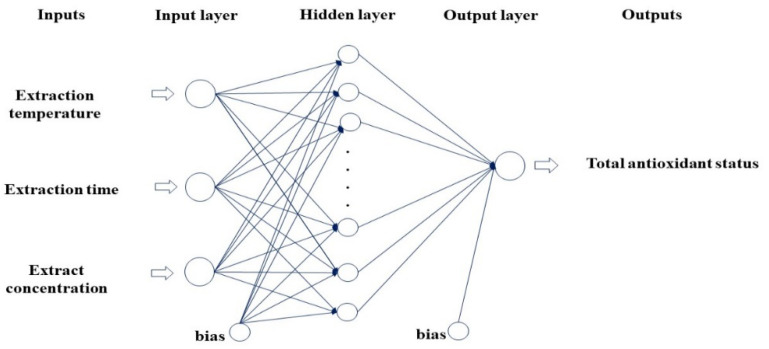
Layers of the ANN model.

**Figure 2 foods-13-01560-f002:**
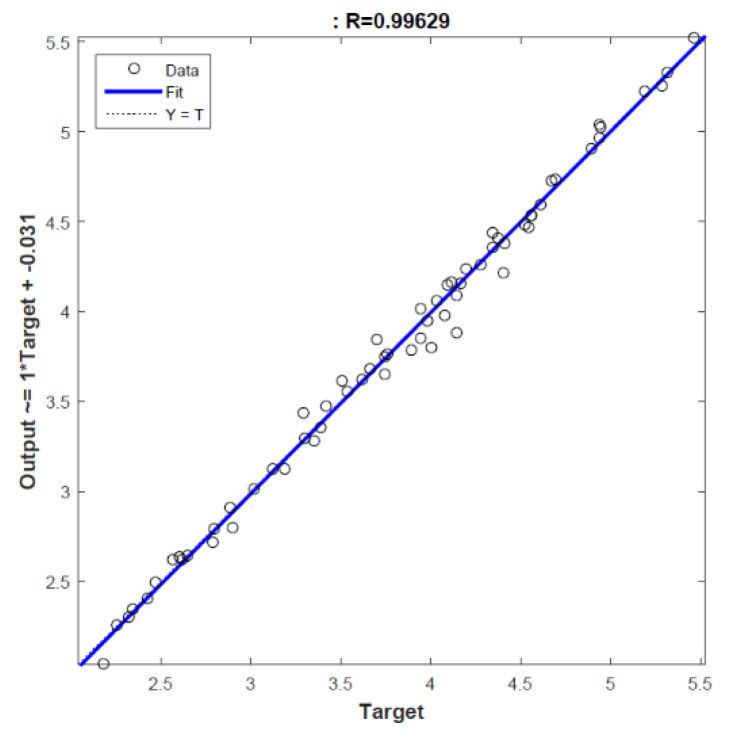
Regression plot of TAS output.

**Figure 3 foods-13-01560-f003:**
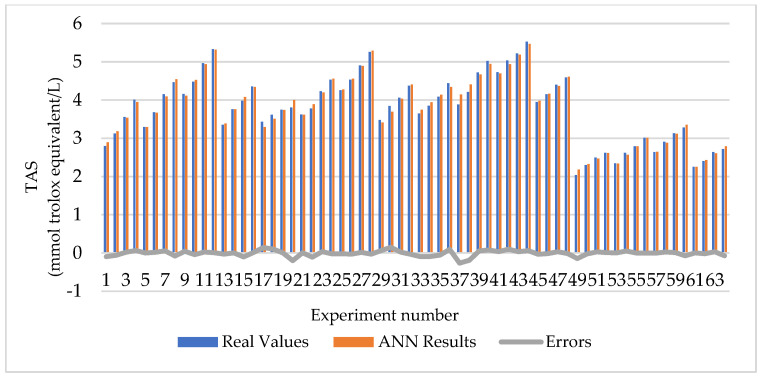
Real values and ANN results for TAS.

**Figure 4 foods-13-01560-f004:**
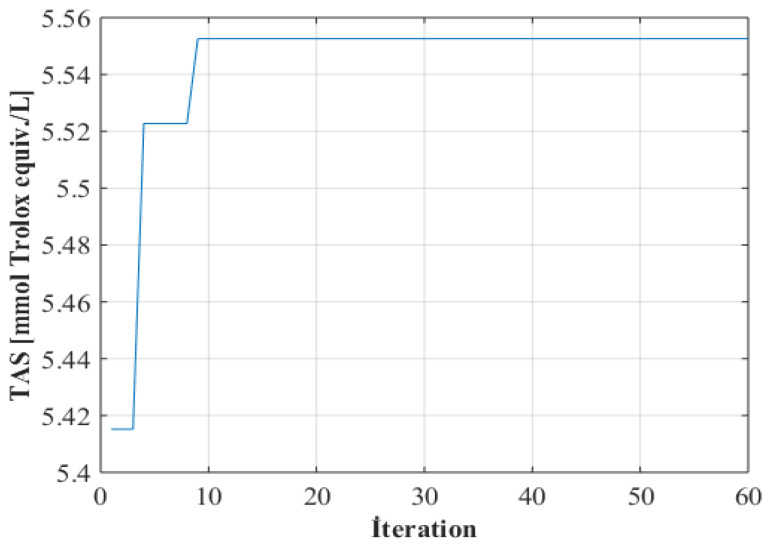
The convergence curve of the optimization process.

**Figure 5 foods-13-01560-f005:**
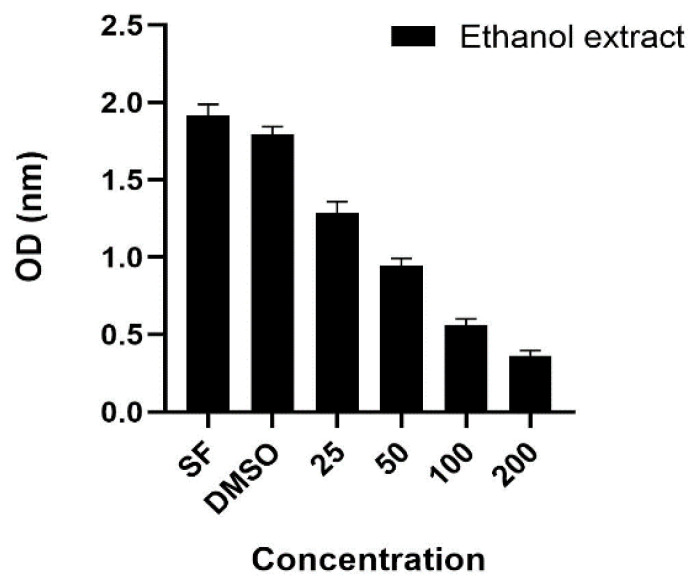
Antiproliferative effect of ethanolic extract of *H. erinaceus*. (DMSO: dimethyl sulfoxide; SF: serum-free medium; OD: nanometer; 25, 50, 100, and 200 μg/mL: extraction concentrations).

**Table 1 foods-13-01560-t001:** Total antioxidant status values of the ethanolic extracts of *H. erinaceus*.

Experiment Number	Extraction Temperature (°C)	Extraction Time (h)	Extract Concentration (mg/mL)	TAS (mmol Trolox Equivalent/L)
1	40	3	0.25	2.799 ± 0.233
2			0.5	3.127 ± 0.121
3			1	3.556 ± 0.074
4			2	4.011 ± 0.173
5		5	0.25	3.297 ± 0.106
6			0.5	3.683 ± 0.115
7			1	4.149 ± 0.119
8			2	4.465 ± 0.212
9		7	0.25	4.159 ± 0.186
10			0.5	4.480 ± 0.141
11			1	4.965 ± 0.076
12			2	5.325 ± 0.125
13		9	0.25	3.355 ± 0.139
14			0.5	3.759 ± 0.115
15			1	3.979 ± 0.086
16			2	4.356 ± 0.205
17	50	3	0.25	3.433 ± 0.088
18			0.5	3.612 ± 0.059
19			1	3.746 ± 0.066
20			2	3.803 ± 0.080
21		5	0.25	3.624 ± 0.033
22			0.5	3.782 ± 0.073
23			1	4.234 ± 0.066
24			2	4.535 ± 0.070
25		7	0.25	4.260 ± 0.072
26			0.5	4.530 ± 0.119
27			1	4.907 ± 0.064
28			2	5.254 ± 0.122
29		9	0.25	3.475 ± 0.049
30			0.5	3.846 ± 0.103
31			1	4.058 ± 0.084
32			2	4.374 ± 0.063
33	60	3	0.25	3.651 ± 0.118
34			0.5	3.851 ± 0.089
35			1	4.086 ± 0.142
36			2	4.439 ± 0.105
37		5	0.25	3.882 ± 0.098
38			0.5	4.214 ± 0.168
39			1	4.725 ± 0.158
40			2	5.020 ± 0.113
41		7	0.25	4.731 ± 0.091
42			0.5	5.035 ± 0.061
43			1	5.221 ± 0.093
44			2	5.323 ± 0.109
45		9	0.25	3.950 ± 0.102
46			0.5	4.152 ± 0.094
47			1	4.404 ± 0.046
48			2	4.594 ± 0.090
49	70	3	0.25	2.038 ± 0.096
50			0.5	2.301 ± 0.107
51			1	2.496 ± 0.059
52			2	2.623 ± 0.118
53		5	0.25	2.344 ± 0.127
54			0.5	2.619 ± 0.089
55			1	2.791 ± 0.059
56			2	3.010 ± 0.119
57		7	0.25	2.643 ± 0.063
58			0.5	2.906 ± 0.095
59			1	3.128 ± 0.051
60			2	3.282 ± 0.074
61		9	0.25	2.256 ± 0.059
62			0.5	2.407 ± 0.064
63			1	2.638 ± 0.092
64			2	2.717 ± 0.110

**Table 2 foods-13-01560-t002:** Anti-AChE and anti-BChE values for ethanolic extract of *H. erinaceus*.

	AChE (μg/mL)	BChE (μg/mL)
Ethanol	13.85 ± 0.94 ^b^*	28.00 ± 0.89 ^b^
Galantamine	6.21 ± 0.62 ^a^	25.01 ± 0.43 ^a^

* Means followed by different letters differ significantly at *p* < 0.05 (Duncan’s multiple-range test).

**Table 3 foods-13-01560-t003:** Minimum inhibitory concentration (MIC) values of ethanolic extract of *H. erinaceus*.

	*S. aureus*	*S. aureus* MRSA	*E. faecalis*	*E. coli*	*P. aeruginosa*	*A. baumannii*	*C. glabrata*	*C. albicans*	*C. krusei*
Ethanol extract	25	25	50	100	100	100	100	50	100
Ampicillin	1.56	3.12	1.56	3.12	3.12	-	-	-	-
Amikacin	-	-	-	1.56	3.12	3.12	-	-	-
Ciprofloksasin	1.56	3.12	1.56	1.56	3.12	3.12	-	-	-
Flukanazol	-	-	-	-	-	-	3.12	3.12	-
Amfoterisin B	-	-	-	-	-	-	3.12	3.12	3.12

Concentrations of 25, 50, 100 µg/mL represent the lowest concentrations that inhibit the growth of microorganisms. Mushroom extract in the range of 12.5–800 μg/mL was tested against microorganisms. Ampicillin, Amikacin, Ciprofloksasin, Flukanazol, Amfoterisin B: standard antibiotics.

**Table 4 foods-13-01560-t004:** Total antioxidant, total oxidant, oxidative stress index, and total phenolic values of ethanolic extract of *H. erinaceus*.

TAS (mmol/L)	5.426 ± 0.123
TOS (µmol/L)	6.621 ± 0.197
OSI (TOS/(TAS × 10))	0.122 ± 0.003
TPC (mg/g)	59.75 ± 1.82
DPPH (mg Trolox equivalent/g)	73.36 ± 2.04
FRAP (mg Trolox equivalent/g)	107.66 ± 2.41

Values are given as mean ± standard deviation (n = 3).

**Table 5 foods-13-01560-t005:** Phenolic contents of ethanolic extract of *H. erinaceus*.

Phenolic Compounds	Values (mg/kg)
Acetohydroxamic acid	2.46415
Catechinhyrate	17.54653
Vanillic acid	<LD
Syringic acid	<LD
Thymoquinone	<LD
Resveratrol	9.4593
Myricetin	1.361
Kaempferol	<LD
Fumaric acid	9.39341
Gallic acid	92.00393
Protocatechuic acid	3.8488
4-hydroxybenzoic acid	0.40874
Caffeic acid	<LD
Salicylic acid	<LD
Phloridzindyhrate	0.23083
2-hydoxycinamic acid	0.94018
Oleuropein	<LD
2-hyroxy1,4 naphthaquinone	<LD
Naringenin	0.42601
Silymarin	<LD
Quercetin	2.34281
Luteolin	0.44483
Alizarin	<LD
Curmin	<LD

## Data Availability

The original contributions presented in the study are included in the article, and further inquiries can be directed to the corresponding author.

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
