# Peer review of "Biological Activities of Ethanol Extracts of Hericium erinaceus Obtained as a Result of Optimization Analysis"

_foods, 2024, doi:10.3390/foods13101560_

Round 1

Reviewer 1 Report

Comments and Suggestions for Authors

In this study, biological activity of ethanol extracts from the wild edible mushroom Hericium erinaceus was determined. In this context, firstly, the extraction conditions were compared: temperature 40-70ËšC, extraction time 3-9 hours and extraction conditions 0.5-2 mg/mL. As a result, extraction conditions were optimized and total antioxidant status (TAS) analysis was. The data obtained from the experimental study was modelled with artificial neural networks (ANN), one of the artificial intelligence methods, and optimized with a genetic algorithm (GA).

The purpose and justification of the work are good. The materials and methods section sounds good. The statistics data should be added to Table 1. The discussion and conclusions are appropriate. opinion, the work requires some corrections.

 Abstract: Results of Total antioxidant status in the optimal extraction conditions should be indicated before reporting the results of the Anti-AChE and the Anti-BChE. Note that the Anti-AChE and the Anti-BChE results were obtained only with extracts obtained under optimal conditions.

Line 18: total antioxidant status

Line 19: TAS instead of tas

Lines 23-28: Is this information needed in the summary or is it sufficient to include it only in the material and methods section?

Lines 33-34: Explain the abbreviation: Anti-AChE the Anti-BChE and add unit to numerical values. Data quoted in the text of the work usually do not provide standard deviations when they are provided in tables.

Material and Method

Abbreviations should be explained at the time of first mention. Explaining the abbreviations in the material and methods section will make reading the work easier.

Tables and Figure: The title and description of the table should be fully informative and contain complete information, so that the reader can understand the data contained in the table without referring to other parts of the work.

e.g. Table 1. Total antioxidant status values of the ethanolic extracts of H. erinaceus.

Add statistical data to Table 1.

Table 2: Means followed by different letter(s) in columns differ significantly at p<0.05

Figure 3. The chart is incomprehensible. I suggest changing the title to: Antiproliferative effect of ethanolic extract of H. erinaceus.

Explain: OD (nm), SF,

Concentration of what? Add units.

Table 3: Explain MIC

Title: Comparison of MICs of H. erinaceus ethanol extracts and antibiotics

Mushroom extract in the ranges of 12.5-800 μg/mL was tested against microorganisms - this information should be included in the Table 3.

Explain: Ampicillin, Amikacin, Ciprofloksasin, Flukanazol, Amfoterisin. Mark clearly in the table that these are the results of other researchers.

Line 102: Explain MTT

Line 108: Was the solution centrifuged? What parameters, device manufacturer?

Line 115: Ellman method [6]   instead of  Ellman method.

Linss 130… Use Italics for Latin names

Line 142: Total antioxidant status and total oxidant status

Line 147: Explain the abbreviation OSI

Lines 164-167: check space between number and units

Lines 165-166: was instead of is

Line 183: at 60ËšC, 7 hours and 2 mg/mL   instead of  at 70ËšC, 3 hours and 0.25 mg/mL

Line 327: Table insead of table

Table 5. Use SI unit

References should be adapted to the requirements of the journal.

For more see attached file.

Author Response

Hi Dear Reviewer.

Thank you very much for your suggestions to improve the article.

All corrections have been made and are included in the attached reply file to the referee.

Best regards.

Reviewer 2 Report

Comments and Suggestions for Authors

The subject of the presented study: “Biological Activities of Ethanol Extracts of Hericium erinaceus Obtained as a Result of Optimization Analysis” and the results obtained are of interest for a readership of the Foods journal.

The article concerns the chemical (content of phenolic compounds, flavonoid content) and biological analysis of ethanol extracts obtained from the wild edible mushroom Hericium erinaceus. The work presents a wide range of biological tests of extracts: antioxidant potential, antimicrobial activity, inhibition of acetyl and butyrylcholinesterase activity and antiproliferative activity. This is an important advantage of the presented research. The data obtained from the experimental study was modeled with artificial neural networks (ANN) and optimized with a genetic algorithm (GA).

The paper is planned correctly and well-written

I think the article needs the following corrections and additions:

Below I present my comments

1. The authors should discuss in more detail the method of preparing biological material for extraction and the procedure for preparing extracts.

2. In section 2.6.: How were the optimal conditions for the determination determined (enzyme, substrate, inhibitor concentration)? No reference number for the Ellman method is provided in section 2.6. The concentration of the enzyme solution used or its activity should also be provided.

3. Authors often use the statement: in our study (e.g. in line 163, 278, 292 and others) - they should use: in this study (or a similar statement).

4. line 252 and below: names of bacterial and fungal strains should be written in italics.

5. Please standardize the unit Trolox/Equivalent: in table 5 there is Trolox/equivalent; Table 1: trolox equivalent; picture 3: Trolox/equiv.; Table 4: Trolox Equi.

6. it is not explained what OD(nm) means in Figure 3

7. in table 2 there is: ethanol, I think it should be: ethanol extract

8. Chapter 3.6: there are no units for DPPH and FRAP (lines: 298 - 303).

Kind regards

Author Response

(The authors gave the same response as above.)

Reviewer 3 Report

Comments and Suggestions for Authors

1.      Line 20: “tas”. It should be uppercase “TAS”.

2.      Introduction. This section should introduce the state of the art in the field. A quick search of the term “Hericium erinaceus” in Scopus reveals a lot of studies about this mushroom, including antioxidant activities. The introduction should focus on related studies.

3.      Line 50-51: “Fruiting bodies …”. What is the relevance of this information for current study?

4.      Lines 55-57: Including or excluding criteria? Size, weight, shape, maturity stage?

5.      Lines 59-61: The authors should indicate that they followed a 4x4x4 or 4^3 experimental design. Replicates?

6.      Section 2.2: Why did you use an ANN model instead of a traditional second order model such as those used in RSM or better yet, a theoretical extraction model, as your data can be arranged as 16 extraction kinetics?

7.      What do you mean by extract concentration? Mushroom-to-solvent ratio?

8.      Lines 59-62. What solvent did you use for extraction experiments? Ethanol is mentioned in lines 99-100.

9.      Soxhlet extraction is performed at the boiling temperature of the solvent. Ethanol has a normal boiling point of 78.37° so how did you achieve the reported extraction temperatures?

10. Did investigated factors have a significant effect on responses?

11. What is the factor having the highest and lowest effect on responses?

12. The problem with ANN regression models is that they might reproduce physically inconsistent trends in within the experimental region. The authors should include surface plots of the predicted response at several combinations of investigated factors, that is, a plot of TAS as a function of extraction temperature and extraction time at a fixed concentration, a plot of TAS as a function of extraction temperature and concentration at a fixed extraction time, and a plot of TAS as a function of extraction time and concentration at a fixed extraction temperature.

13. What is the purpose of using GA to optimize the ANN? Shallow ANNs such as that proposed in the study can be trained with the Solver tool in Excel and the same tool can be used to obtain an unconstrained or constrained optimum value. Do you suspect ANN model has local minima?

14. The optimization problem was not properly stated. It can be anticipated that TAS would increase along mushroom-to-solvent ratio unless saturation of the extraction medium is reached (according to mass transfer theory). Please notice that according to Table 1 the highest TAS was always obtained at 2 mg/mL. In fact, a traditional optimization methodology such as path of steepest ascent would have recommended performing experiments outside of the experimental region.

15. How did you implement experiments at 60.667°C?

16. Lines 189-192. How did you arrive to this conclusion? The methodology does not describe any model selection techniques. I would recommend the Akaike information criterion (AIC) or the AIC corrected for small sample size (AICc).

17. Section 3.1. How do TAS values compare with those from other products?

18. Why did you expect the conditions yielding the highest TAS would be also suitable for the remaining assays?

19. Antimicrobial potential. How are you sure that inhibition was due to mushroom compounds instead of ethanol?

Author Response

(The authors gave the same response as above.)

Reviewer 4 Report

Comments and Suggestions for Authors

The paper entitled „Biological Activities of Ethanol Extracts of Hericium erinaceus Obtained as a Result of Optimization Analyse“ describes determine the phenolic composition and biological activities of extracts of Hericium erinaceus mushroom obtained under  optimum condition.
The topic is interesting, but the paper is superficial and not good structured. The paper require significant improvement before acceptance. I suggest major revision.
Keywords: Antialzeihmer; anticancer; antimicrobial; antioxidant; Hericium erinaceus; medicinal
Mushroom.... Authors should add anticancer activity, antimicrobial activity antioxidant activity, also, authors should add ANN as  keywords.
It was determined that the Anti-AChE value was 13.85±0.94 and the Anti- 33 BChE value was 28.00±0.8. In the abstract, authors should add units for enzyme inhibition assays.
Authors should separate information of samples and sample preparation as a one part. Authors should add more information about samples, samples identification, collection, storage before analysis.
Introduction is superficial.  Authors should add hypothesis of research, aims of the study, overview of recent literature related to the topic and highlight need for research.
Line 52 ...determine the biological activities of extracts of Hericium erinaceus mushroom obtained under  optimum condition. Authors in the paper did chemical and biological characterisation, and mentioned in aims ot the paper just biological. Please correct it.
Authors should mention software in material and methods and add some references related to Eq 1 and 2.
Line 87. The optimization procedure was done using GA. Sentence  is not clear.
Line 162. Phenolic Analysis. Authors should add more details about analysis.
Line 130-133. Names of strains should be italic.
Line 182 and 183.  2.038±0.096 is Add unit.
Figure 2 and 3 could be combined as one.
Table 5. Authors should express phenolic content as µg/g or mg/g not as ppb, None should be replaced as Ë‚LD
 p-coumaric acid…p should be italic, Authors should correct it in all manuscript.

Comments on the Quality of English Language

The authors should improve quality of the English languages.

Author Response

(The authors gave the same response as above.)

Round 2

Reviewer 3 Report

Comments and Suggestions for Authors

1.      Previous comment. The authors were advised that introduction section should introduce the state of the art in the field. Besides, the purpose of the section is to analyze the topic of the study to identify the knowledge gap to be filled. The authors barely modified a rather short section.

2.      Previous comment. The authors were warned that ANN regression models might reproduce physically inconsistent trends within the experimental region. So, they were advised to include surface plots of the predicted response at several combinations of investigated factors. The authors’ response was that “creating surface plots could significantly increase the length of the article and complicate the process of deriving key findings from the model”. This response is unacceptable as 3 plots can be easily grouped in a single figure and occupy a negligible space in the document.

3.      Previous comment. The authors were questioned about the adequacy of using GA to optimize the ANN, specially for such an easy problem. Please notice that the existence of several local minima or a very complex behavior in an ANN model (as stated in one of the authors’ response) for such a limited factor space (3 factors) is indicative of physically inconsistent trends within the experimental region, as warned in the previous comment. Therefore, the use of GA is not justified.

4.      Previous comment.  The authors were warned that it could be anticipated that TAS would increase along mushroom-to-solvent ratio unless saturation of the extraction medium is reached (according to mass transfer theory). Thus, optimization should have extended beyond the initial experimental region. The authors’ response was “It's reasonable to anticipate an increase in TAS along the mushroom-to-solvent ratio. This highlights the limitations of traditional optimization methodologies”. What limitations? The optimum point is clearly outside of the experimental region and traditional methodologies would have clearly highlight this issue, as opposed to GA.  Therefore, optimization problem was not properly defined.

5.      I am not sure if the extraction performed corresponds to a Soxhlet extraction as it is carried out at the boiling point of solvent and the authors report lower temperatures. The only way to reduce temperature in Soxhlet extraction is by reducing pressure, but I think the equipment used by the authors does not have this characteristic. Besides, the manufacturer (Gerhardt) does not refer to their systems as Soxhlet extractors.

Author Response

Hi Dear Rewiewer

I hope you are fine.

Thank you for your valuable feedback. We have further improved our article by taking your suggestions into account. Thank you very much for your support to make the article better.

You can take a look at our answers attached.

Best regards.

Reviewer 4 Report

Comments and Suggestions for Authors

Authors improved manuscript.

Author Response

Hi Dear Reviewer

Thank you for your valuable feedback.

Best regards.